# Evaluation of Indian Mustard Genotypes for White Rust Resistance Using *BjuWRR1*Gene and Their Phenotypic Performance

Yengkhom Sanatombi Devi [1], Th. Renuka Devi [1,*], Ajay Kumar Thakur [2], Umakanta Ngangkham [3], H. Nanita Devi [4], Pramesh Kh. [5], Bireswar Sinha [6], Pusparani Sanjam [7], N. Brajendra Singh [1] and Lokesh Kumar Mishra [8]

1 Department of Genetics and Plant Breeding, College of Agriculture, Central Agricultural University, Imphal 795004, Manipur, India
2 Directorate of Rapeseed-Mustard Research (ICAR), Bharatpur 321303, Rajasthan, India
3 ICAR Research Complex for NEH Region, Manipur Centre, Imphal 795004, Manipur, India
4 All India Co-ordinated Research Programme (Soybean), College of Agriculture, Central Agricultural University, Imphal 795004, Manipur, India
5 All India Co-ordinated Research Programme (Groundnut), College of Agriculture, Central Agricultural University, Imphal 795004, Manipur, India
6 Department of Plant Pathology, College of Agriculture, Central Agricultural University, Imphal 795004, Manipur, India
7 All India Co-ordinated Research Programme-Rapeseed-Mustard, College of Agriculture, Central Agricultural University, Imphal 795004, Manipur, India
8 Department of Biochemistry, Physiology, Microbiology and Environmental Science, College of Agriculture, Central Agricultural University, Imphal 795004, Manipur, India
* Correspondence: renukath139@gmail.com

**Abstract:** The present investigation was carried out to identify the potential donors of resistant gene(s)/the source of white rust disease in *B. juncea* using 30 genotypes, including locally adapted accessions and advanced breeding lines. Out of 30 genotypes, ten lines viz. Bio-YSR, CAULC-1, CAULC-2, CAULC-3, CAULC-4, CAURM-2, CAULR-7, CAURM-4, CAURM 4-1, and CAURM 4-2 exhibited a lower PDI value (lesser than mean 10.83) with a superior agronomic performance related with the disease. The evaluation of these ten genotypes for the presence of the *BjuWRR1* gene using a gene-based marker depicted the presence of the functional allele of the *BjuWRR1* gene in the five genotypes *viz.*, Bio-YSR, CAULC-1, CAULC-3, CAURM 4-1 and CAURM 4-2. When compared with the sequenced amplicon of these genotypes, it is found to be identical with that of an east European *Brassica juncea* line, Donskaja-IV, the completely resistant genotype against various isolates of *Albugo candida*. The findings from the present study suggested that besides Bio-YSR, the local lines of Manipur CAULC-1 (Local Yella of Posangbam) and CAULC-3 (Local Yella of Kakching Lamjao) can be used as the potential white rust resistance sources/donors in disease resistance breeding programmes for the development of elite *B. juncea* cultivars in the future. In addition to the local lines, two improved advanced lines, *viz.* CAURM 4-1 and CAURM 4-2, obtained from a hybridization programme, may be further evaluated for releasing resistant varieties against white rust.

**Keywords:** *Albugo candida*; *BjuWRR1*; *Brassica juncea*; resistant; susceptible

## 1. Introduction

Indian Mustard (*Brassica juncea*), belonging to the rapeseed-mustard group of oilseed crops, is a very economically important oilseed crop in India which is presently being cultivated over an acreage of 6.82 mha [1]. It is also becoming a very popular and profit-making oilseed crop of north-eastern India, particularly in the states of Assam, Manipur and Meghalaya. It is being grown worldwide, especially for its oil, and it is also preferred

as a condiment or as a green leafy vegetable in some parts of the world [2]. However, a number of biotic and abiotic stresses are affecting the overall production and productivity of this crop. Among the biotic stresses, white rust caused by a biotrophic oomycete, *Albugo candida* (Pers.) Kuntze, is a serious disease of Indian mustard [3]. Phenotypically, white rust disease is characterized by the presence of white blister or pustules on cotyledon, the abaxial leaf surface, inflorescence, the base of leaf petiole/stem and the formation of the stag head at the later stage of the plant, if a severe infestation occurs in the plant. The infestation of pathogen prefers a cold night with warm days and the additional soil moisture after it has rained. In India, almost all the commercially released mustard varieties are susceptible to white rust disease [4,5]. The yield loss encountered by *A. candida* in *B. juncea* ranges from 10 to 70% in India, and sometimes it reaches up to 90% depending upon the severity of disease and the environmental conditions prevailing during the season [6]. Additionally, the Indian gene pool of *B. juncea* is highly susceptible to *A. candida* as compared to the east European gene pool [4,5]. As a common disease management practice, use of fungicides has been followed by most of the mustard growers, which ultimately affects the environment. However, it is of a well-known fact that the availability of resistant varieties is one of the cheapest and environmentally friendly options.

The host resistance mechanism is dependent on two defence mechanisms, i.e., first on the cell surface of the host by the recognition of the conserved pathogen-associated molecular patterns (PAMPs) of the invading pathogen, and the second is the recognition of the pathogen-secreted effectors (Avr) via the intercellular receptors encoded by the R gene (s), called (ETI) mechanism or effector-triggered immunity [7]. The resistance R gene of *A. Candida* was earlier reported as a dominant monogenic gene in *B. juncea* [8,9]. Different R genes were studied in the different *Brassica* species *viz. AC-1* [10] and *AC-2* [11]. A single major white rust resistance locus *ACB1-A4.1* was mapped on the linkage group A04 in Heera (the partially resistant east European line), and the *ACB1–A5.1* locus present on the linkage group A05 in Donskaja–IV (fully resistant east European line) of *B. juncea* [4]. Borhan et al. [12,13] reported three loci *RAC1, RAC2* and *RAC3* from two accessions, Ksk-1 and Ksk-2, and *WRR4* from the accession Col of *Arabidopsis thaliana.* The *WRR4* gene was reported to be a broad-spectrum disease resistance gene against various races of pathogen (race1, race2, race4, race7 and race9) in *A. thaliana* and showed a complete resistance against race2 while race7 in *B. juncea* and *B. napus*, respectively [14]. Apart from the *WRR4* gene in *Arabidopsis*, *BjuWRR1* (a constructively expressing gene encoding CC-NB-LRR domain-containing protein) gene provides a complete resistance against *A. candida* and its allelic diverseness in different *Brassica* germplasm was also reported [15]. The *BjuWRR1* gene was identified from a resistance-conferring locus, earlier mapped as *AcB1-A5.1* in the east European gene pool of *B. juncea*-Donskaja-IV. Additionally, the CNL type of the R gene (*BjuA046215/BjuWRR2*) was mapped in the *B. juncea* -Varuna and Turmida $F_1$DH lines [16].

The identification of more stable R gene (s) is a prerequisite practice to improve the disease management efficacy, apart from the white rust-resistant plant varieties that have been developed. The ideal pre-breeding resources may help in *Brassica* improvement programs and potential parents can be used as the donor line for developing durable disease-resistant plant varieties while maintaining the strong performance traits of elite varieties. So, the exploration of the available genetic resources, the evaluation and the exploitation of the promising disease tolerant lines is a realistic alternative and sustainable way to optimize the crop production without using a chemical approach. It is of the fact that traditionally grown primitive cultivars and the wild relatives of cultivated plants are the basic genetic resources that not only sustained the present-day crop improvement but are also required to meet the aspirations of future generations to face the unforeseen challenges of biotic and abiotic stresses. Therefore, the chief objective of this study was to identify the presence of the R gene (*BjuWRR1)* as well as to screen the available Indian mustard genotypes, including the locally adapted genotypes against white rust disease with desirable agronomic traits, which is in close relation with the disease.

## 2. Materials and Methods

### 2.1. Plant Material

The plant material in the present investigation was comprised of 30 genotypes of Indian mustard (24 germplasm accessions/varieties with 6 checks), including 4 local germplasm/accessions collected from different parts of Manipur (CAULC-1, CAULC-2, CAULC-3 and CAULC-4) and 7 fixed advanced lines (CAURM-1, CAURM-2, CAURM-5, CAURM 4-3, CAURM 4-2, CAURM-4 and CAURM-4-1) (Table 1). All the thirty genotypes were screened for the phenotypic expression of white rust disease under natural field conditions during *Rabi* season 2018–19 at the experimental field of the Department of Genetics and Plant Breeding, College of Agriculture, CAU, Imphal, Manipur. The experiment was conducted in a complete augmented block design (ABD) in 4 blocks with 6 check varieties. Each block was spaced by 1 m. The row-to-row and plant-to-plant distances were maintained at 30 cm and 15 cm, respectively. The six check varieties consisted of BIO-YSR and JM-1 as resistant, Basanti and NRCHB-101 as tolerant and Varuna and NRCDR-2 as susceptible, which were planted at random within every block. The recommended agronomic practices like thinning, weeding, fertilizer application, etc., were followed, but fungicides were avoided during the crop season to provide a maximum disease pressure in the natural field condition.

**Table 1.** List of Indian mustard genotypes used in the present investigation along with their source.

| Sl. No. | Genotypes | Source | Sl. No. | Genotypes | Source |
|---|---|---|---|---|---|
| 1. | CAULC-1 | Awang Potshangbam, Manipur | 16. | Pusa Mustard-28 | DRMR, Bharatpur, Rajasthan |
| 2. | CAULC-2 | Kitchenware, Manipur | 17. | Laxmi | DRMR, Bharatpur, Rajasthan |
| 3. | CAULC-3 | KakchingLamjao, Manipur | 18. | Bio-YSR | DRMR, Bharatpur, Rajasthan |
| 4. | CAULC-4 | Sekmai, Manipur | 19. | JD-6 | DRMR, Bharatpur, Rajasthan |
| 5. | CAURM-1 | AICRP (RM) CAU, Imphal Centre | 20. | Urvashi | DRMR, Bharatpur, Rajasthan |
| 6. | CAURM-2 | AICRP (RM) CAU, Imphal Centre | 21. | GM-2 | DRMR, Bharatpur, Rajasthan |
| 7. | CAURM-5 | AICRP (RM) CAU, Imphal Centre | 22. | Rajendra Suflam | DRMR, Bharatpur, Rajasthan |
| 8. | CAURM 4-3 | AICRP (RM) CAU, Imphal Centre | 23. | Pusa Bold | DRMR, Bharatpur, Rajasthan |
| 9. | CAURM 4-2 | AICRP (RM) CAU, Imphal Centre | 24. | RH-30 | DRMR, Bharatpur, Rajasthan |
| 10. | CAULR-7 | AICRP (RM) CAU, Imphal Centre | 25. | JM-1 | DRMR, Bharatpur, Rajasthan |
| 11. | BPR-547 | AICRP (RM) CAU, Imphal Centre | 26. | Basanti | DRMR, Bharatpur, Rajasthan |
| 12. | CAUMC-28 | AICRP (RM) CAU, Imphal Centre | 27. | NRCHB-101 | DRMR, Bharatpur, Rajasthan |
| 13. | CAURM-4 | AICRP (RM) CAU, Imphal Centre | 28. | RH-749 | DRMR, Bharatpur, Rajasthan |
| 14. | CAURM-4-1 | AICRP (RM) CAU, Imphal Centre | 29. | Varuna | DRMR, Bharatpur, Rajasthan |
| 15. | JM-2 | DRMR, Bharatpur, Rajasthan | 30. | NRCDR-2 | DRMR, Bharatpur, Rajasthan |

### 2.2. Disease Reaction

Observations on the occurrence of the disease for analysing the percent disease index (PDI) were taken from 10 randomly selected plants in each line of each block at 8-day intervals during the vegetative as well as true leaf stage, i.e., 42 days after sowing (DAS), 50 DAS, 58 DAS, 66 DAS and 74 DAS under natural conditions. The disease incidence was recorded following a 0–9 scale of Fox and William [17], as follows:

- Zero (immune for WR): no lesions on either cotyledon surface.
- One (HR): non-sporulating pin-point size necrotic spot/a >5% leaf area covered by lesions.
- Three (R): slightly sporulating minute 1–2 mm diameter necrotic spot/5–10% leaf area covered by lesions.
- Five (MR): moderately sporulating small pustules of about a 2–4 mm diameter for larger spots/a 11–25% leaf area covered by the spots.
- Seven (S): moderately sporulating, many large pustules of about a 4–5 mm diameter/a 26–50% leaf area covered by the lesions.
- Nine (HS): profusely sporulating, large coalescing pustules of a > 6 mm diameter/a > 50% leaf area covered by the lesions without margins.

$$\text{Average severity score} = \frac{\sum (n \, X \, 0) + (n \, X \, 1) + (n \, X \, 3) + (n \, X \, 5) + (n \, X \, 7) + (n \, X \, 9)}{Number \; of \; leaf \; samples} \quad (1)$$

$$\text{PDI (\%)} = \frac{\sum (n \, X \, 0) + (n \, X \, 1) + (n \, X \, 3) + (n \, X \, 5) + (n \, X \, 7) + (n \, X \, 9)}{Total \; number \; of \; plants/leaves \; observed \; X \; Maximum \; disease \; score \; (9)} \quad (2)$$

where *n* = the number of leaves in the respective score.

### 2.3. Phenotypic Evaluation

The following observations were recorded to select the better agronomically performing genotype apart from the percent disease index.

- Days to 50% flowering: The number of days counted from the date of sowing to the date of attainment of a 50% flowering was recorded on plot basis.
- Days to 80% maturity: The number of days taken from the date of sowing to 80% siliqua turning yellow was recorded on plot basis.
- 1000 seed weight (g): The mean weight of a 1000 seed weight was recorded from the selected plants.
- Seed yield/Plant (g): The total yield produced by individual plants were recorded from the selected plants.
- Oil content (%): The total oil content of the samples was recorded using a pre-calibrated seed grader machine (FOSS Infratec$^{TM}$1241 Grain Analyzer) of selected plants for each genotype.

Statistical Analysis of the Data

An analysis of variance (ANOVA) and the genetics parameters (PCV-Phenotypic coefficient of variance, GCV-Genotypic coefficient of variance, $h^2$- Heritability at broad sense and GAM-genetic advance as percent mean) of the five phenotypic data were calculated using the R studio software, version 4.2.1.

### 2.4. Physiological Evaluation (Stomatal Density Analysis)

The leaf samples of 45 days old plants were taken by peeling of the leaf from the lower epidermis. The thin slice of epidermis was treated with a drop of saffranine solution, followed by keeping on the slide, and covered with a coverslip for an observation under a microscope. The stomata were observed under a microscope at 40X with the help of Bio wizard 4.2 image analysis software. The average number of stomata in each leaf was counted from three images per samples.

### 2.5. Screening of BjuWRR1 Gene

The plant genomic-DNA was isolated from two-week-old seedling using the CTAB method [18]. The extracted genomic-DNA was quantified using a spectrophotometer. For quality checking, the DNA samples were run on 0.8% agarose gel and thereafter documented the gel using Image Lab Software in the Gel Doc$^{TM}$ XR+ BIO-RAD (Hercules, CA, USA). The concentrated DNA samples were diluted to 100 ng/μL using 1 X TE buffer and stored at −20 °C for further use. A PCR amplification was carried out using gene-specific primers to determine the presence/absence of the *BjuWRR1* gene which is mapped at *AcB1–A5.1* on the locus [15] (Table 2). The PCR reaction was carried out in a 20 μL reaction volume containing 2 μL of 10 × Taq buffer, 2.0 μL of 50 mM MgCl$_2$ solution, 1.5 μL of 2.5 mM dNTPs mixture and 0.5 μL of each of the forward and reverse primers at a concentration of 10 pmole/μL, 0.2 μL of 3 U/μL T*aq* DNA polymerase, 4 μL of diluted genomic DNA (100 ng/μL) and 8.3 μL of nuclease-free water.

**Table 2.** Primer details used in the present investigation.

| Gene | Primer | Sequence (5′-3′) | T$_m$ (°C) | Expected Amplicon | |
| | | | | Varuna (Susceptible) | Donskaja-IV (Resistant) |
| --- | --- | --- | --- | --- | --- |
| *BjuWRR1* | F-DV R-D R-V | GGCATAGTATTCCCTAGAAGAGAGATAAC TGTTGATTCTTAGAATGGTAAATCACAG TTGAAAATCACATGTATACATATGGCTT | 58 | 761 bp | 366 bp |

The PCR was performed using a thermocycler (Applied Biosystems, 2720 Thermal Cycler) with the following cycle: an initiation denaturation at 95 °C for 3 min, followed by 35 cycles of denaturation at 94 °C for 1 min, annealing at 55 °C for 1 min and an extension at 72 °C for 1 min and a final extension at 72 °C for 7 min. The PCR products were separated on a 3% agarose gel stained with ethidium bromide, in a 1X TAE buffer. Thereafter, a gel documentation was followed.

*2.6. Multiple Sequence Alignment of BjuWRR1 Gene*

In order to further confirm the sequence similarity of the *BjuWRR1* gene (encoding CC–NB–LRR-type proteins) sequence from the different genotypes, a partial sequencing of the PCR-amplified regions using primers (*BjuWRR1*) was done using a Sanger sequencing technique. The sequences of five genotypes such as CAULC-1, CAULC-3, CAURM4-1, CAURM 4-2 and BIO-YSR were successfully submitted to NCBI. The assembled gene sequence was annotated with the reference gene sequence of *BjuWRR1* (available in BRAD database) and compared using BLASTn for the nucleotide sequence in the non-redundant (nr) databases of NCBI. The best blast hit was determined from among the score and E-values of the sequences, producing significant alignments [19].

A dendrogram was generated by using the UPGMA clustering method in MEGA Software, Version 11 [20]. The known gene sequence of *A. thaliana* (NM104527.4) and *B. juncea* (MK469977.1) conferring resistance against white rust disease was downloaded from the NCBI or BRAD database. The gene sequence of five genotypes along with *Arabidopsis* and *B. juncea* were subjected to multiple sequence alignment (MSA) using the ClustalW algorithm in BioEdit software (http://www.mbio.ncsu.edu/BioEdit/bioedit.html; accessed on 12 January 2022), followed by the development of dendrogram analysis using MEGA Software.

## 3. Results

*3.1. Disease Reaction and Phenotypic Evaluation*

The phenotypic disease reaction (0–9 scale), collected at 8-day intervals among 30 genotypes of Indian mustard evaluated for a white rust disease reaction under a natural field condition, showed a significant variation (Figures 1 and 2, Table 3). The percent disease index (PDI) of the genotypes calculated from the phenotypically recorded disease reaction ranged from 2.44 to 22.31, with a general mean value of 7.70 (Table 3). A significantly lower PDI value than the general mean as well as the resistant check variety BIO-YSR (9.87) were recorded by the local genotypes collected from various parts of Manipur state, India, such as CAULC-4 (2.44) from Sekmai, followed by CAULC-3 (2.80) from Kakching Lamjao, CAULC-2 (3.73) from Kakching Wairi, CAULC-1 (4.11) from Awang Potsangbam and CAUR-7 (6.42) broad leaf mustard from Imphal. On the other hand, the highest PDI was observed in the genotype Varuna (22.31) (national check susceptible line), followed by RH-30 (21.09). However, no complete immunity against white rust disease was observed among the genotypes studied in the present investigation.

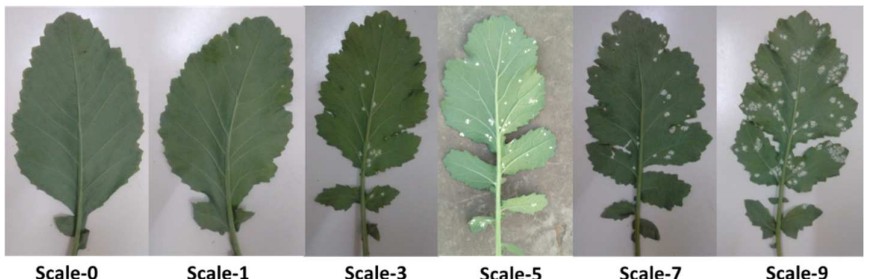

**Figure 1.** Photographic 0–9 scale of disease reaction for screening of genotypes at true leaf stage.

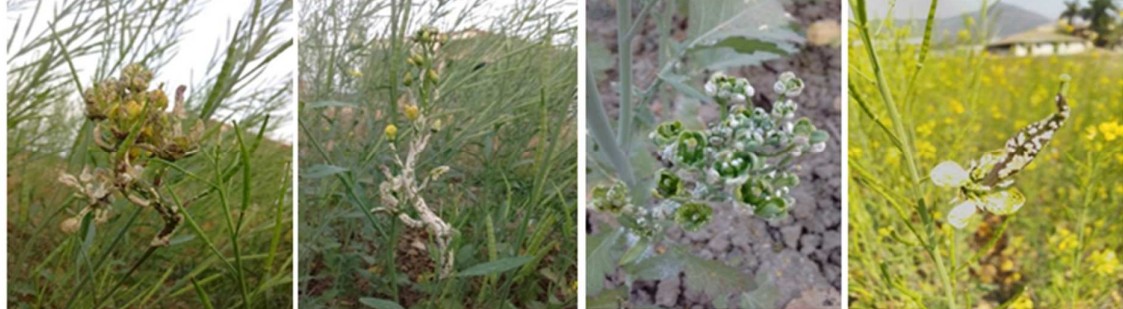

**Figure 2.** Deformed plant parts due to *A. candida* infection.

**Table 3.** Mean performance and genetic parameters of 30 Indian mustard genotypes for various agronomic traits.

| Genotype | 50% DF | 80% DM | Y/P (IN) | Y/P (NIP) | 1000 SW (g) | OC (%) | PDI (%) | Stomata Density/mm² |
|---|---|---|---|---|---|---|---|---|
| G1 (CAULC-1) | 66.21 | 121.80 | 4.62 | 4.74 | 3.26 | 43.84 | 4.11 (2.19) | 146.67 |
| G2 (CAULC-2) | 65.21 | 122.13 | 2.20 | 2.26 | 2.38 | 38.13 | 3.73 (2.11) | 130.67 |
| G3 (CAULC-3) | 67.54 | 123.46 | 2.39 | 2.34 | 2.07 | 40.44 | 2.80 (1.90) | 69.33 |
| G4 (CAULC-4) | 77.37 | 141.63 | 3.06 | 4.04 | 1.65 | 43.96 | 2.44 (1.89) | 74.67 |
| G5 (CAURM-1) | 57.71 | 119.46 | 3.28 | 3.15 | 4.88 | 43.11 | 15.07 (3.78) | 136.00 |
| G6 (CAURM-2) | 59.71 | 120.46 | 3.31 | 3.57 | 4.34 | 45.10 | 7.46 (2.92) | 106.67 |
| G7 (CAURM-5) | 56.71 | 121.46 | 3.87 | 4.44 | 4.77 | 45.45 | 12.95 (3.74) | 69.33 |
| G8 (CAURM 4-3) | 63.37 | 119.12 | 2.82 | 3.85 | 3.28 | 41.11 | 13.15 (3.77) | 138.67 |
| G9 (CAURM 4-2) | 64.71 | 118.79 | 2.93 | 3.52 | 4.48 | 45.31 | 8.64 (3.12) | 85.33 |
| G10 (CAULR-7) | 54.05 | 114.46 | 2.35 | 2.18 | 3.76 | 42.77 | 6.42 (2.34) | 117.33 |
| G11 (BPR-547) | 56.38 | 118.79 | 2.83 | 2.84 | 4.53 | 44.30 | 13.31 (3.54) | 109.33 |
| G12 (CAUMC-28) | 59.37 | 120.30 | 2.97 | 5.39 | 4.84 | 46.16 | 13.95 (3.96) | 130.67 |
| G13 (CAURM-4) | 64.37 | 121.79 | 3.27 | 4.07 | 4.36 | 48.91 | 6.99 (2.84) | 88.00 |
| G14 (CAURM-4-1) | 57.04 | 124.12 | 3.69 | 4.25 | 2.99 | 43.79 | 9.22 (3.21) | 93.33 |
| G15 (JM-2) | 59.54 | 120.46 | 4.06 | 4.38 | 4.45 | 43.29 | 16.55 (4.04) | 109.33 |
| G16 (Pusa Mustard-28) | 46.71 | 113.79 | 3.78 | 6.06 | 3.16 | 43.05 | 11.46 (4.71) | 80.00 |
| G17 (Laxmi) | 60.05 | 128.46 | 4.06 | 6.61 | 3.50 | 43.24 | 13.51 (3.56) | 128.00 |
| G18 (Bio-YSR) | 61.00 | 120.50 | 4.05 | 4.75 | 3.48 | 41.79 | 9.87 (3.17) | 98.67 |
| G19 (JD-6) | 55.87 | 117.80 | 2.67 | 3.79 | 3.49 | 43.62 | 11.91 (3.47) | 120.00 |
| G20 (Urbashi) | 61.54 | 125.46 | 5.17 | 7.81 | 6.92 | 42.40 | 12.38 (3.20) | 85.33 |
| G 21 (GM-2) | 74.71 | 136.63 | 3.82 | 6.58 | 3.92 | 44.05 | 11.24 (3.58) | 82.67 |
| G 22 (Rajendra Suflam) | 64.37 | 123.63 | 3.01 | 6.16 | 4.23 | 42.99 | 12.71 (3.72) | 117.33 |
| G 23 (Pusa Bold) | 66.04 | 122.96 | 4.02 | 5.86 | 3.50 | 43.51 | 12.18 (3.71) | 210.67 |
| G 24 (RH-30) | 62.71 | 127.63 | 3.56 | 7.69 | 4.73 | 43.04 | 21.09 (4.33) | 173.33 |
| G 25 (JM-1) | 64.50 | 120.50 | 3.42 | 4.30 | 3.95 | 42.87 | 11.31 (3.43) | 98.67 |
| G 26 (Basanti) | 60.71 | 121.79 | 3.04 | 3.09 | 3.85 | 43.94 | 14.22 (2.40) | 165.33 |

**Table 3.** *Cont.*

| Genotype | 50% DF | 80% DM | Y/P (IN) | Y/P (NIP) | 1000 SW (g) | OC (%) | PDI (%) | Stomata Density/mm$^2$ |
|---|---|---|---|---|---|---|---|---|
| G 27 (NRCHB-101) | 53.75 | 124.25 | 3.23 | 5.39 | 5.04 | 43.35 | 14.41 (4.06) | 157.33 |
| G 28 (RH-749) | 63.50 | 135.25 | 4.99 | 8.80 | 3.65 | 44.13 | 12.18 (3.81) | 133.33 |
| G 29 (Varuna) | 57.50 | 121.75 | 3.31 | 5.27 | 2.73 | 43.44 | 22.31 (4.78) | 114.67 |
| G 30 (NRCDR-2) | 64.00 | 126.50 | 3.67 | 5.44 | 3.61 | 44.36 | 12.06 (3.54) | 168.00 |
| Mean | 61.54 | 123.30 | 3.51 | 4.75 | 3.86 | 43.54 | 7.70 (3.36) | 117.96 |
| Range | 46.71–77.37 | 113.79 141.63 | 2.2–5.17 | 2.26–8.80 | 1.65–6.92 | 38.1–48.91 | 2.44–22.31 | |
| GCV (%) | 9.99 | 4.71 | 17.44 | 26.29 | 28.12 | 4.15 | 20.38 | |
| PCV (%) | 10.19 | 4.79 | 25.26 | 32.00 | 28.32 | 4.65 | 22.04 | |
| h$^2$ (%) | 95.99 | 96.78 | 47.66 | 67.47 | 98.57 | 79.56 | 85.49 | |
| GAM (%) | 20.18 | 9.56 | 24.84 | 44.54 | 57.59 | 7.63 | 38.87 | |
| SE | 1.917 | 1.616 | 0.981 | 1.325 | 0.200 | 1.398 | 0.431 | |
| C.D. (5%) | 4.086 | 3.445 | - | 2.824 | 0.425 | - | 0.919 | |

Values in parentheses are $\sqrt{x}$ + 0.5 transformed values. Fifty % DF = days to 50% flowering, 80% DM = days to 80% maturity, IN = infected plants, NIP = non-infected plants, SW = seed weight, OC(%) = oil content, PDI(%) = percent disease index, GCV = genotypic coefficient of variation, PCV = phenotypic coefficient of variation, h$^2$ = broad sense heritability, SE = standard error, C.D.= critical difference at 5% level of probability.

Data on the phenotypic traits *viz.*, the days to a 50% flowering, the days to an 80% maturity, the seed yield/plant (g), the 1000 seed weight (g), the oil content (%), using a FOSS Infratec$^{TM}$1241 Grain Analyzer, and the number of stomata/mm$^2$ were recorded to support the selection of promising Indian mustard lines. The analysis of variance (ANOVA) showed a significant variation in most of the traits studied except for the seed yield/plant (infected) and oil content (Table S1). However. the highest seed yield/plant was recorded for the genotype RH-749 (8.80 g in the non-infected and 4.99 g in the infected plants), followed by Urvashi (7.81 g in the non-infected and 5.17 g in the infected plants) and RH-30 (7.69 g in the non-infected and 3.56 g in the infected plants). Out of which, RH-30 showed both a high mean PDI as well as a high seed yield/plant. The genotype PM-28 flowered and matured at the earliest date (46 days and 113 days) as compared to other lines. The highest oil content was found in CAURM-4 (48.91%), followed by CAUMC-28 (46.16%) and CAURM-5 (45.45%).

In general, the phenotypic coefficient of variation (PCV) was found to be higher than the genotypic coefficient variation (GCV) in all the traits considered, but with the least difference between the PCV and GCV. The broad sense heritability was highest for the characters 1000 seed weight (98.57%), followed by the days to an 80% maturity (96.78%), days to a 50% flowering (95.99%) and the PDI (85.49%). A low heritability was observed in the seed yield of the infected plants (47.66%), while a high heritability (98.57%) coupled with s high genetic advance at the percent mean (57.59) was observed only for the character 1000 seed weight.

### 3.2. Stomatal Density with Disease Infection

In plants, *A. candida* enters into the plant host cell via the stomata. In the case of the preformed resistance system, the stomatal density was examined in order to help in identifying the resistant genotypes. In the present study, no systematic pattern of the increase or decrease value was observed for the character stomatal density as compared with the PDI value. However, the genotypes possessing a moderate to higher number of stomatal densities/mm$^2$ showed a higher PDI *viz.*, Varuna (PDI-22.31), RH-30 (PDI-21.09) and NRCDR-2 (PDI- 12.06) possessed 114.67/mm$^2$, 173.33/mm$^2$ and 168/mm$^2$, respectively. While the genotypes CAULC-4 (PDI-2.44) and CAULC-3 (PDI-2.80) possessed 74.67/mm$^2$ and 69.33/mm$^2$, respectively. (Table 4). Genotypes possessing a lower PDI value *viz.*, CAULC-2 (PDI-3.73), CAULC-1 (PDI-2.19) and CAULR-7 (PDI-6.42) showed a high stomatal density (130.67/mm$^2$), (146.67/mm$^2$) and (117.33/mm$^2$), respectively.

**Table 4.** BLASTn result with the query and sequenced amplicon.

| Sl. No | Genotype Name | Collection Place | Amplicon Size (bp) | Query Gene | Accession No. | Query Coverage (%) | Total Score (S) | E-Value | Max. Identity (%) |
|---|---|---|---|---|---|---|---|---|---|
| 1. | CAULC-1 | Manipur | 366 | *BjuWRR1* | OM243099 | 67 | 372 | $2 \times 10^{-98}$ | 94.65 |
| 2. | CAULC-3 | Manipur | 366 | *BjuWRR1* | OM243103 | 97 | 595 | $9 \times 10^{-166}$ | 99.10 |
| 3. | CAULC-4 | Manipur | 366 | *BjuWRR1* | OM243167 | 89 | 532 | $7 \times 10^{-153}$ | 94.33 |
| 4. | CAURM 4-1 | AICRP(R-M), CAU, Imphal Centre | 366 | *BjuWRR1* | OM243101 | 97 | 573 | $4 \times 10^{-159}$ | 97.63 |
| 5. | CAURM 4-2 | AICRP(R-M), CAU, Imphal Centre | 366 | *BjuWRR1* | OM243100 | 96 | 564 | $3 \times 10^{-156}$ | 97.33 |
| 6. | BIO-YSR | DRMR, Bharatpur | 366 | *BjuWRR1* | OM243102 | 92 | 549 | $7 \times 10^{-152}$ | 97.53 |

The S-score is a measure of the similarity of the query to the sequence; the E−value is a measure of the reliability of the S score.

### 3.3. Validation and Screening of BjuWRR1 Gene

The current study used the previously characterized white rust resistance gene *BjuWRR1* which was validated in two check lines, i.e., Varuna (susceptible) and Bio-YSR (resistant), to test the marker efficiency (Figure 3). The amplicon size of both cultivars was similar to the expected amplicon size, as reported in Donskaja-IV [15], which indicated that the selected marker was working to differentiate between the susceptible and resistant lines. In the present study, ten lines exhibited a less than the mean PDI value (10.83) with a desirable phenotypic performance *viz.*, Bio-YSR, CAULC-1 (Awang Potsangbam, Manipur), CAULC-2 (Kakching Wairi, Manipur), CAULC-3 (Kakching Lamjao, Manipur), CAULC-4 (Sekmai, Manipur), CAURM-2, CAULR-7, CAURM-4, CAURM 4-1 (CAULC-1 X Kranti) and CAURM 4-2 (CAULC-1 x Bio-902), which were selected for genotyping using candidate the gene-based marker of the *BjuWRR1*gene. Out of these 10 lines, only five lines *viz.*, Bio-YSR, CAULC-1, CAULC-3, CAURM 4-1 and CAURM 4-2 exhibited the presence of a positive allele of *BjuWRR1*(Figure 4).

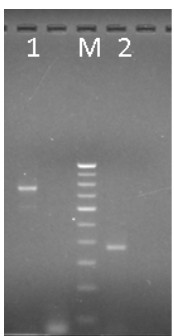

**Figure 3.** Validation of *BjuWRR1*-VD marker: 1-Varuna (761 bp), 2-Bio-YSR (366 bp), M-100 bp ladder.

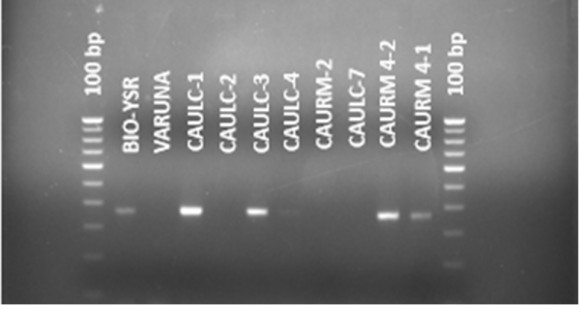

**Figure 4.** Genotyping of 10 *B. juncea* lines showing less than 10.83% PDI (Percent Disease Index) using marker *BjuWRR1*-VD marker.

*3.4. Multiple Sequence Alignment of BjuWRR1 Gene*

In the present study, the sequencing of *BjuWRR1*(using NovaSeq 6000-Illumina) amplicons revealed a high level of similarity between the east European line Donskaja-IV, completely resistance against various isolates of *A. candida* and local lines of Manipur, namely CAULC-1 (Awang Potsangbam, Manipur, India), CAULC-3 (Kakching Lamjao, Manipur, India), CAULC-4, CAURM 4-1, CAURM4-2 and BIO-YSR, respectively (Table S2). The sequence analysis using nucleotide (BLASTn) revealed highest identity with the query sequence was found in CAULC-3 line and followed by CAURM 4-1 and BIO-YSR (Table 4).

The UPGMA (the unweighted pair group method with an arithmetic mean) is a simple and straightforward approach to construct a phylogenetic tree from a distance matrix. Based on a dendrogram, the seven sequence/genotypes separated into two distinct major clusters, I and II (Figure 5A), and amplicon sequences showing the presence of the *BjuWRR1* gene are presented in Table S2. In cluster I, there was only *A. thaliana* indicating the distinctness of the *WRR1 of Arabidopsis* from *B. juncea*. On the other hand, the major cluster II consisted of the remaining six sequences of *B. juncea*. The cluster II was further divided into two sub-clusters (IIA and IIB). The *B. juncea* lines, CAULC 1 in IIA cluster and other such as CAULC-3, CAURM 4-1, CAURM4-2, BIO-YSR and *B. juncea* (MK469977.1) in the IIB sub-cluster. The CAULC-3 and *B. juncea* (MK469977.1) were more similar compared to rest of the genotypes.

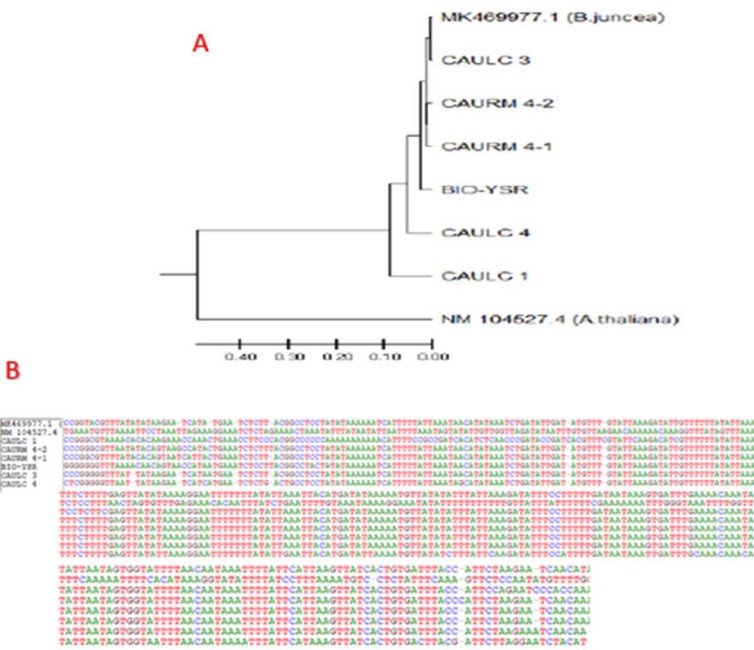

**Figure 5.** (**A**) UPGMA phylogenetic tree generated from MEGA Software. (**B**) Multiple sequence alignment of white rust resistance gene sequence belonging to *B. juncea* and *Arabidopsis thaliana*.

## 4. Discussion

The percent disease index (PDI) of all the 30 genotypes of Indian mustard exhibited a wide variation, ranging from 2.44 to 22.31. The significant differences in the disease infection rate at the same environmental condition among the 30 genotypes of *B. juncea* indicated a different inherent potentiality of each genotype. The genotypes collected from various parts of Manipur, India *viz.*, CAULC-4 from Sekmai, exhibited the lowest PDI value (2.44), followed by CAULC-3 (2.80) from Kakching Lamjao, CAULC-2 (3.73) from Kakching Wairi, CAULC-1 (4.11) from Awang Potsangbam and CAULR-7 (6.42) from Imphal, in comparison to the national check variety Bio YSR (9.87). This may be due the genetic mechanism of the local genotypes which have been grown for a long time and co-adapted with the environment. Previously, several researchers [9,21,22] have also used the same national resistant check line (BIO-YSR) in their studies on the genetics of white rust

resistance. However, complete immunity was not found in the genotypes studied. Similar results were also reported in 74 Indian mustard (*B. juncea*) germplasm lines for a resistance against white rust disease where none of the genotype was found to be resistant [23]. Awasthi et al. [5] reported that almost all the important varieties of *B. juncea* being grown in India were susceptible to white rust.

The highest grain yield/plant in RH-749, Urvashi and RH-30 with a high mean PDI value may be due to their genetic mechanism to reduce the disease infection. The phenotypic coefficient of variation (PCV) was found to be higher than the genotypic coefficient variation (GCV) in all the traits considered. However, the minimum differences between PCV and GCV indicated little environmental effect, influencing the expression of these characters.

The high magnitude of the heritability for the 1000 seed weight (98.57%) coupled with high genetic advances at the percent mean (57.59) indicated that the character is governed by an additive gene action and, therefore, a direct selection could be performed. Similar results of a high heritability for the seed weight (93.68) were also reported by Yadav et al. (2020) [24]. The highest oil content was found in CAURM-4 (48.91%), followed by CAUMC-28 (46.16%) and CAURM-5 (45.45%), which inferred the use of these genotypes for trait-specific breeding.

In plants, *A. candida* enters into the plant's host cell via the stomata. Conversely, the plant's cell has several mechanisms to protect themselves from the pathogen infection. Preformed resistance (the thickness of cuticle and cell wall, the presence of antimicrobial chemicals and enzyme inhibitors, etc.) is one of the disease defence mechanisms, besides an induced resistance system. In the case of the preformed resistance system, the stomatal density was examined in order to help in identifying the resistant genotypes. Two locally collected germplasm lines viz. CAULC-4 (PDI-2.44) and CAULC-3 (PDI-2.80) possessed 74.67 mm$^2$ and 69.33/mm$^2$ stomatal densities, respectively, thus conferring those lower stomatal densities contributed to the disease resistance. The results of Tateda et al. [25] and Dutton et al. [26] reported that less stomatal density on the leaves shows less disease infection, which may be due to a greater number of stomata providing a primary entry point for the pathogen. However, the line CAULC-2, CAULC-1 and CAULR-7 showed a high stomatal density (130.67/mm$^2$), (146.67/mm$^2$) and (117.33/mm$^2$), with a low PDI value (3.73), (4.11) and (6.42), which may be due to certain physiological/biochemical defence mechanisms.

Based on the above results, it has been observed that the genotypes collected from the Manipur region exhibited more tolerance to the white rust pathogen than the other commercial varieties of Indian mustard under the natural conditions of Manipur. This may be due to the adaptation of these genotypes as these have been cultivated in Manipur for so many years, or the presence of a different candidate gene other than *BjuWRR1* located at the *Ac. B1-A5.1* locus [15].

The molecular markers that are tightly linked to the desired trait help to select the better parents at any stage of the plant [27]. Out of the 10 selected genotypes for candidate gene validation, only five lines viz., Bio-YSR, CAULC-1, CAULC-3, CAURM 4-1 and CAURM 4-2, exhibited the presence of a positive allele of *BjuWRR1.* These genotypes may be employed as the potential donors of the *BjuWRR1*gene in white rust disease resistance breeding programmes. The remaining five lines, which are still expressing a resistance reaction, but do not contain a *BjuWRR1* gene such as CAULC-2, CAULC-4, CAURM-2, CAULR-7 and CAURM-4 would be useful for the identification of novel gene(s) for white rust disease resistance. The major disease resistance protein including TIR-NB-LRR, CC-NB-LRR, CC-NBS-LRR, etc. were identified against *A. candida* in several Brassicaceae species [12,13,15,16].

The sequence analysis showed the highest identity with the query sequence found in the CAULC-3 line, CAURM 4-1 and BIO-YSR. The UPGMA divided all the seven sequence/genotypes into two distinct major clusters (I & II).

## 5. Conclusions

The present study is the first report of evaluating different *B. juncea* genotypes against white rust disease in north-eastern India using a candidate gene-based approach. Under natural epiphytotic conditions, the Indian mustard genotypes mostly from Manipur (five local and two fixed advanced lines) exhibited a resistance to *A. candida*, out of which five were found to have the already characterized gene responsible for white rust resistance in an east European germplasm accession of Indian mustard, i.e., Donskaja-IV. The other five genotypes which were not possessing this gene might have another gene encoding for this resistance mechanism which needs to be explored using genetic mapping studies. The white rust resistant Indian mustard genotypes identified in the present study (Bio-YSR, CAULC-1, CAULC-3, CAURM 4-1 and CAURM 4-2) may be employed as donors in white rust breeding programmes and/or further evaluated for releasing as varieties conferring a resistance against white rust in future research endeavours.

**Supplementary Materials:** The following supporting information can be downloaded at: https://www.mdpi.com/article/10.3390/agronomy12123122/s1, Table S1: ANOVA (Analysis of Variance) using Augmented Block Design; Table S2: List of amplicon sequences showing the presence of *BjuWRR1* gene.

**Author Contributions:** Conceptualization and research planning, T.R.D.; data curation, data organization, formal analysis, investigation, methodology, writing—original draft. Y.S.D., T.R.D., U.N. and B.S.; methodology and software T.R.D., B.S., Y.S.D. and U.N.; supervision, T.R.D.; original draft, Y.S.D. and T.R.D.; review and editing, P.K., L.K.M., A.K.T., Y.S.D., T.R.D., U.N., N.B.S., B.S., H.N.D. and P.S. All authors have read and agreed to the published version of the manuscript.

**Funding:** This research received no external funding.

**Data Availability Statement:** Data are contained within the article.

**Acknowledgments:** The authors thank the College of Agriculture, Central Agricultural University, Imphal, Manipur, India for providing all the facilities to carry out the research investigation. We also thank the Director, ICAR-DRMR, Bharatpur, Rajasthan, India for providing seeds of susceptible and resistant Indian mustard lines.

**Conflicts of Interest:** The authors declare no conflict of interest.

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
