# Peer review of "Evaluation of Indian Mustard Genotypes for White Rust Resistance Using BjuWRR1Gene and Their Phenotypic Performance"

_agronomy, doi:10.3390/agronomy12123122_

Round 1
Reviewer 1 Report
Summary:
In this study, Devi et al., investigated the white rust resistance traits of 30 genotypes of Indian mustard. Additionally, they amplified 10 lines with excellent agronomic performance for white rust disease screening with BjuWRR1 gene alleles. They create a phylogenetic tree using the six BjuWRR1 gene alleles that were amplified and sequenced to determine the evolutionary relationship between their lines and European B. juncea. The white rust agronomic resistance phenotypic component of the manuscript is quite comprehensive and simple to comprehend. To enhance their molecular experiments portion, participants could conduct a few more experiments. I'll recommend a major revision to this manuscript overall. And here are my suggestions:
Major suggestions:
1. In Fig. 4, line CAULC-4 has a light band of predicted size, indicating that it has the BjuWRR1 gene. I therefore advise the authors to sequence and purify that band. Additionally, incorporate it into the subsequent studies, including blast and phylogenetic tree.
2. Although CAURM4-1 and CAURM4-2 are present in the paper, only CAURM4.1 is shown in Fig. 4. Please make sure that CAURM4-1 and CAURM4-2 are both included in the gel figure (Fig. 4) and that their names remain the same as in the paper.
Minor suggestions:
3. Since leaf of Scale-3 in Fig. 1 resembles leaf of Scale-1, please swap it out for a different leaf high-quality image.
4. Table S1 with the results of the ANOVA analysis is indicated on line 226. However, neither the website nor the manuscript had this table. Please upload it or enclose it with the manuscript.
5. Please replace the low-quality photographs in Fig. 2 with images of higher quality.
6. Could you please list the sizes of all the amplicons in Table 5 and their sequences in the supplemental table?
7. Fig. 5's quality is also very poor; re-upload a Fig. 5 of higher quality.
8. Verify that the alignment of the sequences is shown in Fig. 5B completely. I believe the authors only displayed a portion of the sequence alignment in the low quality Fig. 5B.
Author Response
Authors’ response to Reviewer #1 comments:
S.No. |
Reviewer’s comments |
Authors’ response |
In this study, Devi et al., investigated the white rust resistance traits of 30 genotypes of Indian mustard. Additionally, they amplified 10 lines with excellent agronomic performance for white rust disease screening with BjuWRR1 gene alleles. They create a phylogenetic tree using the six BjuWRR1 gene alleles that were amplified and sequenced to determine the evolutionary relationship between their lines and European B. juncea. The white rust agronomic resistance phenotypic component of the manuscript is quite comprehensive and simple to comprehend. To enhance their molecular experiments portion, participants could conduct a few more experiments. I'll recommend a major revision to this manuscript overall. And here are my suggestions: |
||
Major suggestions: |
|
|
1 |
In Fig. 4, line CAULC-4 has a light band of predicted size, indicating that it has the BjuWRR1 gene. I therefore advise the authors to sequence and purify that band. Additionally, incorporate it into the subsequent studies, including blast and phylogenetic tree. |
Thank you for your suggestions. The light/faint band of the line CAULC-4 is not considered as the same is not within the acceptable level. |
2 |
Although CAURM4-1 and CAURM4-2 are present in the paper, only CAURM4.1 is shown in Fig. Please make sure that CAURM4-1 and CAURM4-2 are both included in the gel figure (Fig. 4) and that their names remain the same as in the paper. |
Thank you so much for pointing out this mistake in Fig. 4 that CAURM 4-2 is not included in the figure. Actually, the line CAULC1 X BIO 902 in the gel picture Fig.4 have been designated as CAURM 4-2 and now it is replaced as suggested. Even in materials and methods also, the entry G9 (CAULC1 X BIO 902) is replaced by CAURM 4-2 and G8 (CAULC1 X GM-2) by CAURM 4-3.
|
Minor suggestions: |
|
|
1 |
Since leaf of Scale-3 in Fig. 1 resembles leaf of Scale-1, please swap it out for a different leaf high-quality image. |
A high quality image has been provide in the revised manuscript. |
2 |
Table S1 with the results of the ANOVA analysis is indicated on line 226. However, neither the website nor the manuscript had this table. Please upload it or enclose it with the manuscript. |
Table S1 has been enclosed along with the revised manuscript for further processing into the format of the journal. |
3 |
Please replace the low-quality photographs in Fig. 2 with images of higher quality. |
High quality images in Fig. 2 has been incorporated in the revised manuscript. |
4 |
Could you please list the sizes of all the amplicons in Table 5 and their sequences in the supplemental table? |
Thank you for your suggestion, I have added the sizes of all the amplicons in Table 5 and I have also added a separate list of sequences in the supplementary Table (S2).
|
5 |
Fig. 5's quality is also very poor; re-upload a Fig. 5 of higher quality. |
A new figure of high quality has been incorporated in the revised manuscript. |
6 |
Verify that the alignment of the sequences is shown in Fig. 5B completely. I believe the authors only displayed a portion of the sequence alignment in the low quality Fig. 5B. |
I have added a table including all the sequences Shown in Fig.5B |

Reviewer 2 Report
There is no statistical data analysis in the Materials and Methods section.
Table 2 should be deleted as weather conditions are not relevant to the research topic.
L.179-180. What equipment did you use for sequencing?
Table 4. There is no statistical analysis of the data.
Spell out “TW” in the first place.
Did the infection affect the yield? There is no statistical analysis.
The table title says IN while the footer says NI. Check, please.
L.251-255. For this statement, it is necessary to carry out a correlation analysis. For example, Varuna and G10 have almost the same stomatal density (117 and 115), but the PDI differs by almost 4 times (22.3 and 6.4).
Fig. 5B. The picture quality is very poor.
L.325-326. This statement is not supported by statistical analysis.
L.37, 215. A. candida should be italicized.
L.37, 273. B. juncea should be italicized.
L.120-121. In reality, the intervals are 8 days.
L.121. Spell out “DAS” in the first place.
Author Response
Authors’ response to Reviewer #2 comments:
S.No. |
Reviewer’s comments |
Authors’ response |
1 |
There is no statistical data analysis in the Materials and Methods section. |
This portion has been incorporated in the revised manuscript. |
2 |
Table 2 should be deleted as weather conditions are not relevant to the research topic. |
Table 2 has been deleted in the revised manuscript. |
3 |
L.179-180. What equipment did you use for sequencing? |
NovaSeq 6000-Illumina was used for the sequencing purpose. This point has been incorporated in the revised manuscript. |
4 |
Table 4. There is no statistical analysis of the data. |
The data has been statistically analyzed using Analysis of variance (ANOVA) as mentioned in the text. Various parameters like mean, range, GCV, PCV, h2 and GAM have been computed. |
5 |
Spell out “TW” in the first place. |
‘TW’ has been spelled out fully in the first place in the revised manuscript. |
6 |
Did the infection affect the yield? There is no statistical analysis. |
Yes, the infection affected the yield, however, no statistical analysis could be carried out. |
7 |
The table title says IN while the footer says NI. Check, please. |
This point has been corrected accordingly. |
8 |
Fig. 5B. The picture quality is very poor. |
The picture quality of Fig. 5B has been improved. |
9 |
L.325-326. This statement is not supported by statistical analysis. |
We agree with the Reviewer’s opinion. |
10 |
L.37, 215. A. candida should be italicized.
|
A. candida has been italicized throughout the revised manuscript. |
11 |
L.37, 273. B. juncea should be italicized. |
B. juncea has been italicized throughout the revised manuscript. |
12 |
L.120-121. In reality, the intervals are 8 days. |
This point has been corrected in the revised manuscript. |
13 |
L.121. Spell out “DAS” in the first place.
|
‘DAS’ has been spelled out in full form in the first place in the revised manuscript. |
Round 2
Reviewer 1 Report
The author didn't respond favorably to my #1 major suggestion.
CAULC-4 has the lowest PDI in their data, which is why I recommended them to purify and sequence it. Since they cited "Out of these 10 lines, only five lines viz., Bio-YSR, CAULC-1, CAULC-3, CAURM 4-1 and CAURM 4-2 exhibited the presence of positive allele of BjuWRR1(Figure 4)" in lines 268 and 360, adding the CAULC-4's data will also give them more credibility and make their results match those in Fig. 4.
They won't have a hard time carrying out this experiment. Based on their Fig. 4, they had already amplified the BjuWRR gene in CAULC-4. Additionally, the CAULC-4 band is quite distinct (it's visible), making it simple to purify the PCR product by increasing the PCR reaction volume.
Author Response
S.No. |
Reviewer’s comments |
Authors’ response |
1 |
The author didn't respond favorably to my #1 major suggestion. CAULC-4 has the lowest PDI in their data, which is why I recommended them to purify and sequence it. Since they cited "Out of these 10 lines, only five lines viz., Bio-YSR, CAULC-1, CAULC-3, CAURM 4-1 and CAURM 4-2 exhibited the presence of positive allele of BjuWRR1 (Figure 4)" in lines 268 and 360, adding the CAULC-4's data will also give them more credibility and make their results match those in Fig. 4. They won't have a hard time carrying out this experiment. Based on their Fig. 4, they had already amplified the BjuWRR gene in CAULC-4. Additionally, the CAULC-4 band is quite distinct (it's visible), making it simple to purify the PCR product by increasing the PCR reaction volume. |
The suggested work has been done in our lab and the results have been incorporated in the revised manuscript accordingly. |
Reviewer 2 Report
The manuscript has been improved over the original version. However, Table 4 lacks the results of the analysis of independent groups and therefore it is impossible to determine the significance of differences between mustard genotypes for various agronomic traits. In addition, the table numbering needs to be corrected, since Table 2 has been deleted. It is necessary to correct the reference “S1” to “Table S1” (L. 229) and add a reference to Table S2 in the text.
Author Response
S.No. |
Reviewer’s comments |
Authors’ response |
1 |
The manuscript has been improved over the original version. However, Table 4 lacks the results of the analysis of independent groups and therefore it is impossible to determine the significance of differences between mustard genotypes for various agronomic traits. In addition, the table numbering needs to be corrected, since Table 2 has been deleted. It is necessary to correct the reference “S1” to “Table S1” (L. 229) and add a reference to Table S2 in the text.
|
Kindly refer Table S1 (Supplementary table 1) for the result of significant differences between mustard genotypes for various agronomic traits and Table 3 for genetic parameters (Mean, Range, Genotypic Co-efficient Variance, Phenotypic Co-efficient Variance, Heritability and Genetic advance along with Standard Error and Critical Difference at 5 % Level of Probability). Necessary correction for Table S1 and S2 has been added in the text. (Table S1 and S2 enclosed as separate file).
|

Round 3
Reviewer 1 Report
Please also add CAULC-4 in Fig. 5A and 5B.
Author Response
Sequence analysis part of CAULC-4 fragment does not seem to be very much reliable. However, at this stage, this does not seem possible for us to resequence it again due to some official issues. Hence, cannot be incorporated in Table 4 and in dendrogram.